# A tool for the cheap and rapid screening of SARS-CoV-2 variants of concern (VoCs) by Sanger sequencing

Germán Burgos,[1,2] Andrés Ambuludí,[3] USFQ SARS-CoV-2 Consortium,[4] Diana Morales-Jadán,[2] Miguel Angel Garcia-Bereguiain,[2] Claire Muslin,[2] Vinicio Armijos-Jaramillo[3,5]

**ABSTRACT** Severe acute respiratory syndrome coronavirus 2 (SARS-CoV-2) is an emerging virus that, since March 2020, has been responsible for a global and ongoing pandemic. Its rapid spread over the past nearly 3 years has caused novel variants to arise. To monitor the circulation and emergence of SARS-CoV-2 variants, surveillance systems based on nucleotide mutations are required. In this regard, we searched in the spike, ORF8, and nucleocapsid genes to detect variable sites among SARS-CoV-2 variants. We describe polymorphic genetic regions that enable us to differentiate between the Alpha, Beta, Gamma, Delta, and Omicron variants of concern (VoCs). We found 21 relevant mutations, 13 of which are unique for Omicron lineages BA.1/BA.1.1, BA.2, BA.3, BA.4, and BA.5. This genetic profile enables the discrimination between VoCs using only four reverse transcription PCR fragments and Sanger sequencing, offering a cheaper and faster alternative to whole-genome sequencing for SARS-CoV-2 surveillance.

**IMPORTANCE** Our work describes a new (Sanger sequencing-based) screening methodology for SARS-CoV-2, performing PCR amplifications of a few target regions to detect diagnostic mutations between virus variants. Using the methodology developed in this work, we were able to discriminate between the following VoCs: Alpha, Beta, Gamma, Delta, and Omicron (BA.1/BA.1.1, BA.2, BA.3, BA.4, and BA.5). This becomes important, especially in low-income countries where current methodologies like next-generation sequencing have prohibitive costs. Furthermore, rapid detection would allow sanitary authorities to take rapid measures to limit the spread of the virus and therefore reduce the probability of new virus dispersion. With this methodological approach, 13 previously unreported diagnostic mutations among several Omicron lineages were found.

**KEYWORDS** SARS-CoV-2, variant of concern (VoC), Sanger sequencing, receptor-binding domain, SARS-CoV-2 surveillance, Omicron lineages

I n late December 2019, a wave of patients with pneumonia-like symptoms caused great alarm in China due to the unknown source of the illness. Sequencing of samples from those patients revealed that the cause was a novel coronavirus, which was first named 2019-nCoV and is now known as severe acute respiratory syndrome coronavirus 2 (SARS-CoV-2). In March 2020, the novel coronavirus outbreak was designated as a pandemic by the World Health Organization (WHO) (1, 2). In April of the same year, more than 12,000 genomic sequences were annotated on the Global Initiative on Sharing All Influenza Data (GISAID) database (3). Single-nucleotide polymorphisms (SNPs) are the primary type of genetic diversity found in SARS-CoV-2 (4).

A high mutation rate, typical of RNA viruses like SARS-CoV-2, especially in the spike (*S*) gene, has enabled the rapid and concomitant appearance of several lineages (5). As the number of viral variants increased, different systems for classification were proposed,

Address correspondence to Germán Burgos, hgermanburgosf@yahoo.com, or Vinicio Armijos-Jaramillo, vinicio.armijos@udla.edu.ec.

The authors declare no conflict of interest.

See the funding table on p. 6.

the Pango nomenclature being one of them (https://www.pango.network/). In addition, the scientific community agreed to use Greek letters to facilitate variant identification and location on the subsequent phylogenetic scale. Some of these variants were later named variants of concern (VoCs) by the WHO (6). For a lineage to be considered as such, it must be associated with a higher transmission rate, immune evasion ability, decreased effectivity of current therapies and/or detection techniques, and an increment in the number of hospitalizations and deaths. Since the pandemic began, several lineages have been designated as VoCs: the Alpha (B.1.1.7), Beta (B.1.351), and Gamma (P.1) variants in December 2020; Delta (B.1.617.2) in June 2021; and, most recently, the Omicron variant (B.1.1.529, BA.1, BA.1.1, BA.2, BA.3, BA.4, and BA.5 lineages) in November 2021 (7).

The SARS-CoV-2 S protein contains the receptor-binding domain (RBD), a region in which mutations are directly linked to the emergence of new variants (8) and whose function is to interact with the viral receptor ACE2 (angiotensin-converting enzyme 2) in humans (9, 10). Genetic diversity in this viral domain has a variety of implications on virus phenotype, including increased ACE2 affinity, increased intracellular reproduction rate, decreased humoral neutralization efficacy, and even potential bias in diagnostic procedures (e.g., dropout) (11–13). Because of these characteristics, the RBD has been a priority target for drug development and mutation dynamics research (14–19). On the other hand, mutations in the nucleocapsid (*N*) gene have been related to false negatives in quantitative reverse transcription PCR (qRT-PCR) tests, and a greater infectivity capacity (20, 21). *ORF8* is another region of interest; this accessory protein is known to promote immune evasion due to downregulation in MHC-I molecules and interferon beta (22, 23). Although there is a huge effort to sequence and monitor the SARS-CoV-2 genetic variants, developing proper surveillance programs and protocols is challenging for middle- and low-income countries (24, 25). Latin America lacks a large-scale surveillance system to extensively describe virus transmission dynamics due to high costs, in particular for whole-genome sequencing (WGS), and a lack of specialized facilities, making high-sample throughput handling a seriously limiting factor (26–28).

In this work, we describe the analysis of crucial regions within the SARS-CoV-2 genome and the design of a variant identification method using the gold-standard Sanger sequencing approach. We propose a panel of 21 SNPs to discriminate between SARS-CoV-2 Alpha, Beta, Gamma, Delta, and Omicron variants. Given that our methodology can detect VoC-defining mutations, it constitutes, in our opinion, a cost-effective alternative to WGS to make SARS-CoV-2 genotyping and surveillance faster and less expensive, which is especially relevant in a low-income country.

## RESULTS AND DISCUSSION

A multiple sequence alignment of *S*, *ORF8*, and *N* genes extracted from 883 SARS-CoV-2 genomes belonging to the VoCs Alpha, Beta, Gamma, Delta, and Omicron BA.1/BA.1.1, BA.2, BA.3, BA.4, and BA.5 was achieved. To detect variant-specific mutations, a total of 21 relevant polymorphisms were identified. SNP profiles can be observed in Fig. 1. The different Omicron lineages were chosen in order to detect variants with high prevalence and/or increased immune evasion capabilities (29–32) Furthermore, the WHO has designated two of these (BA.4 and BA.5) as "omicron subvariants under monitoring," given their enhanced transmission (6).

As shown in Fig. 1, 13 distinct mutations unique to the Omicron lineages BA.1/BA.1.1, BA.2, BA.3, BA.4, and/or BA.5 were identified. These novels Omicron mutations were named with the prefix OMI followed by a number regarding each one's position along the genome. Eleven of those mutations (OMI1 to OMI11) are located in the RBD region. Three mutations (OMI2, OMI6, and OMI7), were found to be common to the entire Omicron lineage (see mutations highlighted in purple in Fig. 1). These enable discrimination from the other lineages (Alpha, Beta, Gamma, and Delta). The other 10 mutations enable discrimination between the descendant lineages Omicron BA.1/BA.1.1, BA.2, BA.3, BA.4, and BA.5.

| Mutations | | | | | | | | | | | | | | | | | | | | | |
|---|---|---|---|---|---|---|---|---|---|---|---|---|---|---|---|---|---|---|---|---|---|
| **Mutation Name** | T19R | D80A | OMI1 | OMI2 | OMI3 | OMI4 | K417T | OMI5 | OMI6 | T478K | E484K | OMI7 | OMI8 | OMI9 | OMI10 | N501Y | OMI11 | S12 | T716I | OMI12 | OMI13 |
| **Fragment** | D80A | | KS | | | | | | | | | | | | | | | | T716I | PCR35 | |
| **Location** | C21618G | A21801C | T22673C | C22674T | A22688G | G22775A | A22812C | T22917G | G22992A | C22995A | G23012A | A23013C | T23018G | A23040G | G23048A | A23063T | C23202A | A23403G | C23709T | C27889T | C28724T |
| **Variant** | | | | | | | | | | | | | | | | | | | | | |
| **Index:Wuhan:A** | C | A | T | C | A | G | A | T | G | C | G | A | T | A | G | A | C | **A** | C | C | C |
| **Alpha: B.1.1.7** | C | A | T | C | A | G | A | T | G | C | G | A | T | A | G | T | C | G | **T** | C | C |
| **Beta: B.1.351** | C | **C** | T | C | A | G | A | T | G | C | **A** | A | T | A | G | T | C | G | C | C | C |
| **Gamm: P.1** | C | A | T | C | A | G | **C** | T | G | C | **A** | A | T | A | G | T | C | G | C | C | C |
| **Delta: B.1.617.2** | **G** | A | T | C | A | G | A | **G** | G | **A** | G | A | T | A | G | A | C | G | C | C | C |
| **Omicron BA.1/BA.1.1** | C | A | **C** | **T** | A | G | A | T | **A** | **A** | G | **C** | T | G | **A** | **T** | **A** | G | C | C | C |
| **BA.2** | **T** | A | T | **T** | **G** | **A** | A | T | **A** | **A** | G | **C** | T | G | G | T | C | G | C | C | C |
| **BA.3** | C | A | T | **T** | A | **A** | A | T | **A** | **A** | G | **C** | T | G | G | T | C | G | C | C | C |
| **BA.4** | **T** | A | T | **T** | **G** | **A** | A | **G** | **A** | **A** | G | **C** | **G** | **A** | G | T | C | G | C | C | **T** |
| **BA.5** | **T** | A | T | **T** | **G** | **A** | A | **G** | **A** | **A** | G | **C** | **G** | **A** | G | T | C | G | C | **T** | C |

**FIG 1** SNP genetic profiles identified in SARS-CoV-2 variants. The names of previously described mutations are indicated in orange, while the new Omicron-defining mutations detected and named in this study are prefixed OMI and indicated in purple. Those mutations shared within all Omicron lineages are highlighted in purple inside the table. Mutations useful for variant discrimination are indicated by red letters highlighted in lilac.

The other eight relevant mutations observed in this work had already been described by Gangavarapu et al. (30) and Hodcroft (33) and were considered for the discrimination of VoCs (see the orange highlights in Fig. 1). The following paragraph discusses each one.

K417T (A22812C) for the identification of the Gamma variant is involved in protein conformation change and immune response escape (34). T478K (C22995A) was at first distinctive of the Delta variant until the Omicron variant emerged, causing an increment in ACE2 affinity and therefore a reduction of antibody-mediated neutralization (35–38). E484K (G23012A), found in Beta and Gamma variants, increases receptor affinity as well as immune evasion (39, 40). The mutation N501Y (A23063T) that enhances ACE2-Spike protein interaction (41, 42) is present in all VoCs except for Delta. S12 (A23403G) is unique to the index (Wuhan) genome (NCBI Reference Sequence ID: NC_045512.2) (43). T19R (unique for Delta), D80A (unique for Beta), and T716I (unique for Alpha) are involved in changes in transmissibility and neutralization efficacy (5). These already-characterized mutations are present in variants being monitored, such as Mu and Zeta (7, 30, 31). It is important to mention that, given the variability of the amplified regions, we expect to detect more relevant mutations in future VoCs (16, 44–47).

Sequences of 30 qRT-PCR-positive SARS-CoV-2 RNA samples were obtained to test the aforementioned SNP genetic profiles. The genotypes of the samples used for validation are shown in Table S1 in the supplemental material, associated with the access codes of the sequences that detail the diagnostic mutations (see an example in Fig. S1) deposited in GenBank. For each sample, the designated lineages were confirmed with WGS, obtaining a 100% agreement that validates our methodology. Hence, the proposed methodology is suitable for VoC discrimination and is especially useful for studying the Omicron lineage and its descendant lineages under monitoring (BA.4 and BA.5) (6, 30).

Nevertheless, it is likely that, when including more samples, some sequences within a variant present a different SNP profile than the ones described in this study. Hence, we

recommend increasing the number of processed samples to assess the existence of other SNPs and the prevalence of those presented here.

PCR-based genotyping relies on specific mutations that can differentiate between variants. However, detecting the complete set of naturally occurring mutations is difficult, especially in highly polymorphic organisms like SARS-CoV-2. Even though the regions that confer important changes for the establishment and spread of new VoCs were analyzed, it is evident that if a diagnostic mutation occurs in another gene location outside of the considered fragments, our method could not detect it, such as surveillance using WGS does. Moreover, as the approach does not capture every diagnostic mutation found in the SARS-CoV-2 genome, our methodology may fail to identify certain mutations shared among variants or unique to a particular variant. As a result, this method may encounter challenges in detecting new variants. However, it offers a rapid means of identifying samples with known fixed mutations in established variants and a chance to analyze and include new portions of the virus RNA to genotype more mutations. Consequently, it may allow for the identification of potential future variants that would require only slight modifications to the current methodology. In conclusion, this methodology does not replace the detection of new variants using WGS, but it can be valuable for screening and monitoring well-defined variants.

The approach presented in this study can be used to identify lineage-defining SNPs for previous and upcoming VoCs. Unlike other studies where partial or complete genomes are sequenced and many fragments must be amplified, the current method uses only four RT-PCR fragments in a multiplex assay, making the process faster, cheaper, and easier to include in routine analyses (48, 49). At local costs, Sanger sequencing is at least five times cheaper than the WGS alternative (13.8 vs 75.0 USD per sample), even though the turnover time takes almost twice as long, given that individual sequence reads must be done for each sample (up to 20 h vs 10.5 h for WGS). The costs may be lower, depending on the number of readings, and this will depend on the set of diagnostic mutations that are required according to the prevalent or interesting VoC. It should be noted that if resources are not a limitation, the characteristics of the WGS allow it to scale to 96 samples in a processing time not much longer than described. As stated by Bezerra et al. (50), this feature could reduce the gap in genomic surveillance between developing and developed countries.

## Conclusions

We offer a rapid and reliable Sanger sequencing-based method for identifying previous and recent SARS-CoV-2 variants of concern. The finding of new mutations inside the fragments of interest within *S*, *ORF8*, and *N* genes means that the methodology could be used to identify future variants and/or relevant mutations. Three of the 13 SNPs were found to be present across all Omicron lineages presented in this study; however, the combination of some clusters allows discrimination between Omicron descendant lineages. This kind of *in silico* approximation has been shown to accurately anticipate significant genetic changes for SARS-CoV-2 variant categorization. This method can be utilized in healthcare facilities as a low-cost means of handling a lot of samples to speed up virus surveillance.

## MATERIALS AND METHODS

### Identification of unique mutations of SARS-CoV-2 variants

To detect the most prevalent mutations in each variant, a random sample of 880 complete SARS-CoV-2 genomes available in GISAID (https://www.gisaid.org/, accessed 24 April 2022) was downloaded. The combined count of sequences derived from the different variants was as follows: 22 (Alpha), 39 (Beta), 19 (Gamma), 31 (Delta), and 374 (Omicron BA.1/BA.1.1), 100 (Omicron BA.2), 95 (Omicron BA.3), 100 (Omicron BA.4), and 100 (Omicron BA.5). We randomly sampled genomes within each VoC, but we selected

a larger number of samples specifically for the widely distributed Omicron variants that were prevalent during the sampling period. Please note that VoCs BA.3-5 had a relatively low number of sequences available at that time. Once the sequences were obtained, the *S*, *ORF8*, and *N* genes were then extracted using the Geneious Prime 2021.2.2 software (https://www.geneious.com). Genes were aligned with the MAFFT V7 software (using FFT-NS-2 and PAM200) (51). The exclusive and shared mutations among variants were identified manually within each gene sequence.

## Primer design

To amplify the gene fragments that contained the relevant mutations for variant discrimination, several primer pairs were designed with a Tm restriction of 59–61°C using the primer BLAST tool (https://www.ncbi.nlm.nih.gov/tools/primer-blast/) and the Wuhan index virus sequence (NCBI reference sequence ID: NC_045512.2) as a reference. To avoid primer-dimer and hairpin events, primers were analyzed with Autodimer v1 software (52). The selected primer pairs amplified three different fragments of 693, 891, and 370 bp, respectively. Additionally, a primer pair (PCR35) published by Paden et al. (53) was used to amplify another 1,021-bp fragment. Primer sequences and their positions are listed in Table 1.

## Sample collection

The leftovers of 16 qRT-PCR positive SARS-CoV-2 RNA samples for the CDC protocol *N1* and *N2* gene targets (average Ct values for *N1* and *N2* were 19.08 and 19.12, respectively, meaning a viral load of $\geq 10^5$ copies/µL of RNA elution) collected between January and June 2022 were provided by the COVID-19 diagnostic service at Universidad de Las Américas (UDLA) in Quito, Ecuador. Additionally, 14 viral cDNA samples already sequenced via WGS (one wild type, two Alpha, two Gamma, two Delta, two Omicron BA.2, one Omicron BA.3, two Omicron BA.4, and two Omicron BA.5) were supplied by the COVID-19 genomic surveillance project at the Microbiology Institute of Universidad San Francisco de Quito also in Quito, Ecuador.

## Two-step RT-PCR

For cDNA synthesis, the SuperScript III Reverse Transcriptase Kit (Thermo Scientific) was used according to the manufacturer's protocol: 0.5 µM of specific primer KS-REV (Table 1) and 0.25 µM of oligo-dT primers. Then, the amplification of each fragment was performed with the Qiagen Multiplex PCR Kit 1X (QIAGEN, Hilden, Germany) in a total 10-µL multiplex reaction containing 0.2 µM of each primer and 1 µL of the cDNA ($\geq 10^5$ molecules/µL), using Q solution, following the manufacturer's recommendations. The thermal cycling conditions included 15 min at 95°C for enzyme activation. Thermal cycling proceeded with 35 cycles of 94°C for 1 min, 55°C for 1 min 30 s, and 72°C for 1 min. Then, the reaction was quenched at 70°C for 15 min. Amplicons were sized in a 2% agarose gel.

**TABLE 1** Primer sequences for analyzed amplicons

| Amplicon | Size (bp) | Primer | Sequence (5′–3′) | Position (nt)[a] | Gene | Reference |
|---|---|---|---|---|---|---|
| | | D-FWD | AGGGGTACTGCTGTTATGTCTTT | 21421 | | |
| D80A | 693 | D-REV | CCCTGTTTTCCTTCAAGGTCCA | 22113 | | This study |
| | | KS-FWD | TGGAACAGGAAGAGAATCAGCA | 22619 | S | |
| KS | 891 | KS-REV | ACAGCCTGCACGTGTTTGAA | 23509 | | This study |
| | | T-FWD | GTGTGACATACCCATTGGTGC | 23545 | | |
| T716I | 370 | T-REV | TTGTGCAAAAACTTCTTGGGTGT | 23914 | | This study |
| | | 35F1 | ATCTTTTGGTTCTCACTTGAACTGC | 27834 | | |
| PCR35 | 1,021 | 35R2 | TGAACTGTTGCGACTACGTGATG | 28855 | ORF8/N | (53) |

[a]The nucleotide position is described according to the index virus genome sequence (NC_045512.2).

## Sample sequencing

PCR products were purified using Exo1 and FastAP enzymes (Thermo Scientific). The BigDye-Terminator Version 3.1 Cycle Sequencing Kit (Applied Biosystems, Waltham) was used following the manufacturer's recommendations. Sequences were purified by size exclusion chromatography using Sephadex GS50 in the Centri-Sep system (Princeton Separations, Freehold). Sanger sequences were obtained using Data Collection Software v3.3 on the ABI3500 Genetic Analyzer (Applied Biosystems). The quality of the readings was evaluated in the Sequencing Analysis Software (Thermo Scientific). Trimmed reads were imported into Geneious R11-2017 software (Biomatters Ltd.). For each isolate, the sequence reads were mapped to the reference genome (NC_045512.2) where the positions of interest were previously annotated (see Fig. S1). In this way, diagnostic mutations can be quickly identified by SNP calling to assign to the appropriate VoC, following the Pango nomenclature (54).

To confirm the Sanger results, RNAs provided by the COVID-19 diagnostic service at UDLA were also sequenced by WGS in collaboration with the Microbiology Institute of Universidad San Francisco de Quito. Sequencing was performed using GridION from Oxford Nanopore Technologies according to reference (55). Clade and lineage assignments were made by Nextclade (56) and verified on the Pangolin COVID-19 Lineage Assigner platform (57).

## ACKNOWLEDGMENTS

We are immensely grateful to Maria José Muñoz for the continuous accompaniment to the review and discussion of the work in its experimental phase and to Katherine Barrionuevo and Domenyka Hidalgo for their tireless help in reviewing the final sequence files, organized in understandable tables.

USFQ-SARS-CoV-2 Consortium: Mateo Carvajal, Erika Muñoz, Rommel Guevara, Belén Prado, Sully Marquez, Josefina Coloma, Michelle Grunauer, Gabriel Trueba, Verónica Barragán, Patricio Rojas-Silva, and Paúl Cárdenas.

## AUTHOR AFFILIATIONS

[1]Facultad de Medicina, Universidad de Las Américas (UDLA), Quito, Ecuador
[2]One Health Research Group, Faculty of Health Sciences, Universidad de Las Américas (UDLA), Quito, Ecuador
[3]Carrera de Ingeniería en Biotecnología, Facultad de Ingenierías y Ciencias Aplicadas, Universidad de Las Américas (UDLA), Quito, Ecuador
[4]Instituto de Microbiología, Colegio de Ciencias Biológicas y Ambientales (COCIBA), Universidad San Francisco de Quito (USFQ), Cumbaya, Ecuador
[5]Grupo de Bio-Quimioinformática, Universidad de Las Américas (UDLA), Quito, Ecuador

## AUTHOR ORCIDs

Germán Burgos http://orcid.org/0000-0002-8400-2753
Diana Morales-Jadán http://orcid.org/0000-0002-2013-4132
Miguel Angel Garcia-Bereguiain http://orcid.org/0000-0003-0025-3609
Claire Muslin http://orcid.org/0000-0002-2746-6914
Vinicio Armijos-Jaramillo http://orcid.org/0000-0003-2965-2515

## FUNDING

| Funder | Grant(s) | Author(s) |
|---|---|---|
| Universidad de Las Américas Ecuador (UDLA) | MED.GBF.20.06 | Germán Burgos |
| Universidad de Las Américas Ecuador (UDLA) | MED.GBF.20.06 | Andrés Ambuludí |
| Universidad de Las Américas Ecuador (UDLA) | MED.NK.20.10 | Claire Muslin |

| Funder | Grant(s) | Author(s) |
| --- | --- | --- |
| Universidad de Las Américas Ecuador (UDLA) | BIO.TPA.20.03 | Vinicio Armijos-Jaramillo |
| United States Agency for International Development (USAID) | 2022001 | USFQ SARS-CoV-2 Consortium |
| HHS \| NIH \| OSC \| Common Fund (NIH Common Fund) | 2022001 | USFQ SARS-CoV-2 Consortium |
| Red para el desarrollo de instrumentos innovadores aplicados a la investigación epidemiologica en América Latina | 2022001 | USFQ SARS-CoV-2 Consortium |

## AUTHOR CONTRIBUTIONS

Germán Burgos, Conceptualization, Data curation, Formal analysis, Funding acquisition, Investigation, Methodology, Project administration, Resources, Supervision, Validation, Visualization, Writing – original draft, Writing – review and editing | Andrés Ambuludí, Data curation, Formal analysis, Investigation, Methodology, Software, Validation, Visualization, Writing – original draft, Writing – review and editing | USFQ SARS-CoV-2 Consortium, Data curation, Formal analysis, Investigation, Resources, Validation, Writing – review and editing | Diana Morales-Jadán, Data curation, Investigation, Validation, Writing – review and editing | Miguel Angel Garcia-Bereguiain, Data curation, Formal analysis, Investigation, Resources, Validation, Writing – review and editing | Claire Muslin, Data curation, Formal analysis, Investigation, Methodology, Resources, Validation, Visualization, Writing – original draft, Writing – review and editing | Vinicio Armijos-Jaramillo, Data curation, Formal analysis, Investigation, Methodology, Resources, Software, Supervision, Validation, Visualization, Writing – original draft, Writing – review and editing

## DATA AVAILABILITY

GenBank ID codes: OQ976987-OQ976992, OQ978563-OQ978591, OQ978594-OQ978646, OR054008-OR054013, OR073395-OR073400, OR083676-OR083682, OR141938-OR141940, OQ978220-OQ978221, OR077469, OR143349. GISAID Identifier: EPI_SET_230615ky. All sequences in this data set are compared relative to hCoV-19/Wuhan/WIV04/2019 (WIV04), the official reference sequence employed by GISAID (EPI_ISL_402124).

## ETHICS APPROVAL

The severe acute respiratory syndrome coronavirus 2 (SARS-CoV-2) positive samples used in this study were leftovers from samples collected for routine SARS-CoV-2 diagnosis. Nevertheless, this work is included in a study that was approved by the institutional review board of the Dirección Nacional de Inteligencia de la Salud (the Ecuadorian Public Health Ministry's National Health Intelligence Directorate) under code number 008-2020.

## ADDITIONAL FILES

The following material is available online.

### Supplemental Material

**Supplemental Information (Spectrum05064-22-s0001.pdf).** Fig S1: Chromatograms that show ~80bp long genetic profiles (between OMI6 to N501Y) of four samples. Table S1: Diagnostic Mutations of samples used for validation respect to the Wuhan reference sequence (NC_045512.2).

Open Peer Review

**PEER REVIEW HISTORY (review-history.pdf).** An accounting of the reviewer comments and feedback.

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
