## [Reviewer comments · Microbiology Spectrum]

Microbiology Spectrum

A tool for the cheap and rapid screening of SARS-CoV-2 variants of concern (VoC) by Sanger sequencing

Germán Burgos Figueroa, Andrés Ambuludí, The USFQ SARS-CoV-2 Consortium, Diana Morales-Jadán, Miguel Angel Garcia Bereguain, Claire Muslin, and Vinicio Armijos-Jaramillo

Corresponding Author(s): Germán Burgos Figueroa, Universidad de Las Americas

Review Timeline:

Submission Date:	December 9, 2022
Editorial Decision:	April 26, 2023
Revision Received:	June 27, 2023
Accepted:	July 5, 2023

Editor: Eleanor Powell

Reviewer(s): Disclosure of reviewer identity is with reference to reviewer comments included in decision letter(s). The following individuals involved in review of your submission have agreed to reveal their identity: Kenji Ota (Reviewer #1); Pablo Tsukayama (Reviewer #2)

Transaction Report:

DOI: <https://doi.org/10.1128/spectrum.05064-22>

April 26, 2023

Prof. Germán Burgos Figueroa
Universidad de Las Americas
Facultad de Medicina
Campus UdlaPark
Vía Antigua a Nayón
Quito, Pichincha 170503
Ecuador

Re: Spectrum05064-22 (A tool for the cheap and rapid screening of SARS-CoV-2 variants of concern (VoC) by Sanger sequencing)

Dear Prof. Germán Burgos Figueroa:

Thank you for submitting your manuscript to Microbiology Spectrum. After receiving feedback from reviewers, modifications are required before potential publication.

Link Not Available

Sincerely,

Eleanor Powell

Journals Department
Reviewer comments:

Reviewer #1 (Comments for the Author):

1. Whether the method described in this study could be applied to future variants is uncertain. In fact, newly emerging variant can only be identified using WGS. Therefore, depending on this method has potential risk to miss the mutation not included in these fragments. I suggest the authors clearly mention this point as limitation.

2. The variants before the emergence of Omicron have disappeared in clinical strains. Therefore, there seems to be the minimal necessity for discriminating past variants as clinical laboratory testing. On the other hand, this method can be applied for retrospective research purpose.
3. As the authors mentioned, there are intra-variant diversity in the mutations. In silico confirmation of the mutations shown in Table 2 will improve the robustness of this finding.
4. Line 191, I suggest the authors show the result of the sequences of additional 31 samples performed for validation.

Reviewer #2 (Comments for the Author):

This paper reports a technique for screening SARS-CoV-2 variants of concern (VoC) cheaply and quickly using Sanger sequencing. This approach can be used to detect mutations in the virus's spike protein, which is critical for understanding how the virus spreads and how it can respond to antibodies. They also point out that this procedure is straightforward and can be carried out in the majority of molecular biology laboratories. Overall, the study is a valuable resource for researchers and public health laboratories interested in surveillance of SARS-CoV-2 genetic variants.

However, I have some major comments that need addressing:

1. The authors mention that this is a cheaper and faster alternative to WGS. This makes sense intuitively, but, to make the statement, the authors should directly compare both methods' (local) costs and turnaround times. This study could do so since they ran both methods in parallel for a subset of samples.
2. Please provide more details on how the reference set of genomes (n=883) was determined. The authors mention that the selection is random, yet the proportion of VOCs obtained does not look random. What were the selection criteria? Are these from multiple world regions or mainly from South America?
3. On a similar note, given that the study was carried out in Ecuador, the authors could have also included variants of interest Mu (B.1.621) and Lambda (C.37), which originated in South America and reached significant prevalence levels in Ecuador in 2021.
4. By looking at variations across 21 variable sites instead of the entire SC-2 genome, this method is more straightforward than WGS but also suffers from reduced discriminatory power. This and other limitations should be addressed in the discussion.
5. Please provide more details on the analysis of the Sanger sequencing results. Describe the process by which one turns electropherogram data into the SNP genetic profiles in Table 2, so that the readers can reproduce the analysis on their data.

And just one minor comment:

- Revise the text for clarity and flow. In particular, the introduction feels unnecessarily wordy, and the description of the PANGO system could be left out.

Staff Comments:

Preparing Revision Guidelines

Please return the manuscript within 60 days; if you cannot complete the modification within this time period, please contact me. If you do not wish to modify the manuscript and prefer to submit it to another journal, please notify me of your decision

immediately so that the manuscript may be formally withdrawn from consideration by Microbiology Spectrum.

Quito, June 24 /2023

Dr. Eleanor Powell
Editor,
Microbiology Spectrum

Dear Dr. Powell,

Please find enclosed our Response to Reviewers, and supplementary files about the manuscript entitled "*A tool for the cheap and rapid screening of SARS-CoV-2 variants of concern (VoC) by Sanger sequencing*", which was submitted for publication as an Original Research manuscript in *Microbiology Spectrum*. We greatly appreciate both the relevant Reviewers' suggestions received, and the time dedicated by the Microbiology Spectrum editorial team to consider our work as potentially publishable in the journal. After carefully considering the suggestions made by the reviewers, we submit the modified version of our paper for further evaluation. Changes made to the original manuscript submission are highlighted according to your indications.

In the first place, the reviewers agree in suggesting the clarification of some important aspects such as our discriminatory power vs WGS, that could limit the application of our methodology. We found these comments very appropriate because they allowed us to specify our limitations and approach a more realistic estimation of our possible error rate. But beyond this, to significantly enrich our point of view about its potential if we consider that the screening of future variants would require only slight changes to do so. The costs of our method, already lower than those of WGS, could be further reduced if fewer Sanger reads are required, depending on the genomic position of diagnostic mutations that would allow discrimination of cocirculating variants in a specific time frame.

On the other hand, we are grateful for the reviewers' effort and sharpness to improve the understanding of the text by recommending us to better specify certain methodological aspects regarding the sampling for the bioinformatic analysis performed of the different VoCs and the pertinent discrimination analyses that gave us a concrete idea of our possible error rate. A detailed description of the Sanger electropherogram analysis methodology was made regarding the genotypes obtained, which have been consigned in Supplementary Table 1, linked to the GenBank accession codes of the diagnostic mutations, as well as the data (GISAIID) of the samples processed by WGS. We have also included Supplementary Figure 1 as an example of the annotations in a specific portion that allows the discrimination of 4 different VoCs.

We should point out that due to an unintentional editing error in the initial manuscript, we detected some inconsistencies in Table 1 during the subsequent revision process, which are noted as changes and do not consequently affect the discrimination between the VOCs considered, the general considerations regarding the methodology nor the conclusions of the work. We still can discriminate between the VoCs: Alpha, Beta, Gamma, Delta, and Omicron (BA.1/BA.1.1, BA.2, BA.3, BA.4, BA.5).

We sincerely hope that our final considerations meet the expectations suggested for a better understanding of the intention of our work, as a model to search for inexpensive alternatives that can be applied in a larger number of laboratories in order to perform effective surveillance of pathogen distribution in countries where access to public resources is severely limited.

We confirm that this manuscript has not been published elsewhere and is not under consideration by another journal. All authors have approved the manuscript, which is in concordance with *Microbiology Spectrum* guidelines. The study was supported by the MED.GBF.20.06, MED.NK.20.10 and BIO.TPA.20.03 grants from Universidad de Las Américas, Quito-Ecuador and the United States Agency for International Development (USAID), Common Fund (NIH Common Fund), and the Red para el desarrollo de instrumentos innovadores aplicados a la investigación epidemiológica en América Latina for The USFQ SARS-CoV-2 Consortium 2022001. The authors have no conflicts of interest to declare.

Please address all correspondence to:

hgermanburgosf@yahoo.com/german.burgos@udla.edu.ec; Universidad de Las Américas, campus UdlaPark, Quito-Ecuador; Tel +593 23970000 ext 838

vinicio.armijos@udla.edu.ec; Universidad de Las Américas, campus UdlaPark, Quito-Ecuador; Tel +593 23970000 ext 2520

We look forward to hearing from you at your earliest convenience.

Yours sincerely,

Germán Burgos Figueroa MSc. Ph.D. (c), on behalf of all authors.

Response to Reviewers Spectrum05064-22 (A tool for the cheap and rapid screening of SARS-CoV-2 variants of concern (VoC) by Sanger sequencing)

Reviewer comments:

Reviewer #1 (Comments for the Author):

1. Whether the method described in this study could be applied to future variants is uncertain. In fact, newly emerging variant can only be identified using WGS. Therefore, depending on this method has potential risk to miss the mutation not included in these fragments. I suggest the authors clearly mention this point as limitation.

Agreed. In concordance, we added a new paragraph in the manuscript's discussion section (lines 200-213) to address this point and highlight the limitations.

“PCR-based genotyping relies on specific mutations that can differentiate between variants. However, detecting the complete set of naturally occurring mutations is difficult, especially in highly polymorphic organisms like SARS-CoV-2. Even though the regions that confer important changes for the establishment and spread of new VOCs were analyzed, it is evident that if a diagnostic mutation occurs in another gene location outside of the considered fragments, our method could not detect it, such as surveillance using WGS does. Moreover, as the approach does not capture every diagnostic mutation found in the SARS-CoV-2 genome, our methodology may fail to identify certain mutations shared among variants or unique to a particular variant. As a result, this method may encounter challenges in detecting new variants. However, it offers a rapid means of identifying samples with known fixed mutations in established variants; and offers a chance to analyze and include new portions of the virus RNA to genotype more mutations. Consequently, it may allow for the identification of potential future variants that would require only slight modifications to the current methodology. In conclusion, this methodology does not replace the detection of new variants using WGS, but it can be valuable for screening and monitoring well-defined variants.”

We sincerely thank the reviewer for their valuable suggestion and their commitment to improving the manuscript. By including this new paragraph, we aim to provide a clearer explanation of the limitations of our method and the specific scope it can effectively handle.

2. The variants before the emergence of Omicron have disappeared in clinical strains. Therefore, there seems to be the minimal necessity for discriminating past variants as clinical laboratory testing. On the other hand, this method can be applied for retrospective research purpose.

The method aimed to easily detect new variants of concern (VoCs) by being adaptable. To achieve this, we decided to sequence a highly variable genomic region of SARS-COV-2, known for mutations present in all variants. By examining variants such as Alpha, Beta, or Delta, we identified

the specific mutations that distinguish the Omicron variant from other variants of concern (VoCs). Although the method was not originally designed for retrospective purposes, we validated it by testing the primers on preserved samples previously sequenced using NGS. Remarkably, our method produced consistent results in all the tests conducted, suggesting its potential for retrospective research. We would like to express our gratitude to the reviewer for highlighting this prospective application.

3. As the authors mentioned, there are intra-variant diversity in the mutations. In silico confirmation of the mutations shown in Table 2 will improve the robustness of this finding.

The results presented in Table 2 were obtained by aligning 883 sequences. However, in response to the reviewer's concern, we expanded our dataset by downloading 39,263 complete genomes from GISAID (www.epicov.org/epi3/frontend#5069e6-20/05/2023), associated with the Beta variant. After filtering out incomplete genomes, we analyzed a total of 38,447 sequences. These genomes were aligned with the Wuhan sequence NC_045512, and we investigated the nucleotides at positions 21801 and 23012, which serve as diagnostic mutations for the Beta variant (as indicated in Table 2). By excluding "N" and ambiguous bases at position A21801C, we observed a frequency of 0.0149 nucleotides different from "C" in the Beta sequences. Similarly, for G23012A, the frequency of nucleotides different from "A" was 0.026. When considering the independence of these two mutations, our calculations indicate a 0.03% chance of misidentifying Beta variants using our methodology. We attempted to replicate this calculation for other mutations and variants, but due to the large number of sequences available in GISAID (e.g., 8,130,325 genomes for Omicron samples), it became impractical to perform these calculations accurately. We appreciate the reviewer's concern regarding this issue, as it prompted us to test our results with a larger sample size and estimate a realistic error rate for our method.

4. Line 191, I suggest the authors show the result of the sequences of additional 31 samples performed for validation.

We have added in Table S1 of supplementary information the genetic profiles of the requested samples, as well as an example of the profiles in chromatograms in Fig S1 that illustrate the diagnostic mutations for four different variants. The diagnostic mutations of each sample can be evidenced, and we, therefore, appreciate the reviewer's suggestion.

We have included the following description in line 189:

"The genotypes of the samples used for validation are shown in Supplementary Table 1 (Table S1), associated with the access codes of the sequences that detail the diagnostic mutations (see an example in Fig. S1) deposited in GenBank".

Reviewer #2 (Comments for the Author):

This paper reports a technique for screening SARS-CoV-2 variants of concern (VoC) cheaply and quickly using Sanger sequencing. This approach can be used to detect mutations in the virus's spike protein, which is critical for understanding how the virus spreads and how it can respond to

antibodies. They also point out that this procedure is straightforward and can be carried out in the majority of molecular biology laboratories. Overall, the study is a valuable resource for researchers and public health laboratories interested in surveillance of SARS-CoV-2 genetic variants.

However, I have some major comments that need addressing:

1. The authors mention that this is a cheaper and faster alternative to WGS. This makes sense intuitively, but, to make the statement, the authors should directly compare both methods' (local) costs and turnaround times. This study could do so since they ran both methods in parallel for a subset of samples.

A valuable suggestion from the reviewer, thank you very much for pointing it out. We have calculated the costs and turnover times for our methodology (ABI3500 with 8 capillaries) in a set of 24 samples comparable to the two methods. Mainly due to the cost reduction involved, Sanger sequencing would make surveillance more accessible to health systems.

We have included the following paragraph in line 218:

“At local costs, Sanger sequencing is at least 5 times cheaper than the WGS alternative (13.8 vs 75.0 USD per sample), even though the turnover time takes almost twice as long, given that individual sequence reads must be done for each sample (up to 20 hours vs 10.5 hours for WGS). The costs may be lower depending on the number of readings, and this will depend on the set of diagnostic mutations that are required according to the prevalent or interesting VoC. It should be noted that if resources are not a limitation, the characteristics of the WGS allow it to scale to 96 samples in a processing time not much longer than described.”

2. Please provide more details on how the reference set of genomes (n=883) was determined. The authors mention that the selection is random, yet the proportion of VOCs obtained does not look random. What were the selection criteria? Are these from multiple world regions or mainly from South America?

We appreciate your comment, and we agree that this point warrants a more extensive explanation (see Methodology section line 91):

“We randomly sampled genomes within each VoC, but we selected a larger number of samples specifically for the widely distributed Omicron variants that were prevalent during the sampling period. Please note that VoCs BA.3-5 had a relatively low number of sequences available at that time.”

Our approach involved filtering the genomes based on VoCs, followed by the random selection of isolates from various time periods and locations. We acknowledge that this method may introduce bias since certain periods and places have a significantly larger number of sequenced isolates compared to others. We opted for a simple random sampling approach because a proportional sampling method would have been impractical for our specific objectives. Moreover, the rapid spread of SARS-CoV-2 and human mobility in certain countries allowed us to find sample diversity within the same location. Considering these factors, we deemed it reasonable to employ completely random sampling to acquire our set of genomes.

Regarding the second point raised by the reviewer, it is accurate that we did not select a

proportional sample of VoCs. This decision was influenced by our focus on Omicron variants during the time when the samples were collected. Variants such as Alpha, Beta, Gamma, and even Delta were no longer prevalent at that time. We solely utilized this information to identify the established mutations of these variants and also to distinguish Omicron from other variants.

Our data collection encompassed downloads from different countries worldwide. Except for Delta, where most isolates originated from Brazil, the isolates were sourced from diverse countries across the globe, with a notable emphasis on Europe and North America, as well as representation from Africa, Asia, and Australia.

3. On a similar note, given that the study was carried out in Ecuador, the authors could have also included variants of interest Mu (B.1.621) and Lambda (C.37), which originated in South America and reached significant prevalence levels in Ecuador in 2021.

While we acknowledge the observation, our primary focus in this work lies on VoCs, and Mu and Lambda do not attain the same level of importance. Nonetheless, with some additional effort, the methodology can be adjusted to identify other variants, since some of the diagnostic mutations described for the Vols Mu (B.1.621) and Lambda (C.37) are localized in the genomic region analyzed in this work.

The Lambda variant had limited significance in Ecuador, with only 103 (3%) samples detected between September 2021 and May 2022, as reported by GISAID (www.epicov.org). A similar situation was observed for the Mu variant, with 285 (8.3%) samples detected during the same period. Nevertheless, the method can be modified to identify specific mutations in Mu and Lambda for retrospective research.

4. By looking at variations across 21 variable sites instead of the entire SC-2 genome, this method is more straightforward than WGS but also suffers from reduced discriminatory power. This and other limitations should be addressed in the discussion.

We fully agree with the reviewer and appreciate their effort to enhance our manuscript. In response to their feedback, we have added a new paragraph in the manuscript (lines 200-213) discussing the limitations of our method. This includes acknowledging that our method does not substitute whole genome sequencing (WGS) for detecting new variants. Furthermore, we highlight that the sensitivity of our methodology depends on established lineages with fixed mutations.

“PCR-based genotyping relies on specific mutations that can differentiate between variants. However, detecting the complete set of naturally occurring mutations is difficult, especially in highly polymorphic organisms like SARS-CoV-2. Even though the regions that confer important changes for the establishment and spread of new VOCs were analyzed, it is evident that if a diagnostic mutation occurs in another gene location outside of the considered fragments, our method could not detect it, such as surveillance using WGS does. Moreover, as the approach does not capture every diagnostic mutation found in the SARS-CoV-2 genome, our methodology may fail to identify certain mutations shared among variants or unique to a particular variant. As a result, this method may encounter challenges in detecting new variants. However, it offers a rapid means of identifying samples with known fixed mutations in established variants; and offers a chance to analyze and include new portions of the virus RNA to genotype more mutations. Consequently, it may allow for the identification of potential future variants that would require

only slight modifications to the current methodology. In conclusion, this methodology does not replace the detection of new variants using WGS, but it can be valuable for screening and monitoring well-defined variants.”

5. Please provide more details on the analysis of the Sanger sequencing results. Describe the process by which one turns electropherogram data into the SNP genetic profiles in Table 2, so that the readers can reproduce the analysis on their data.

We have modified the paragraph in the methodology of the manuscript (line 131):

“Sanger sequences were obtained using Data Collection Software v3.3 on the ABI3500 Genetic Analyzer (Applied Biosystems). The quality of the readings was evaluated in the Sequencing Analysis Software (Thermo Scientific). Trimmed reads were imported into Geneious R11-2017 software (Biomatters Ltd.). For each isolate, the sequence reads were mapped to the reference genome (NC_045512.2) where the positions of interest were previously annotated (See Fig S1). In this way, diagnostic mutations can be quickly identified by SNP calling to assign to the appropriate VoC.”

We appreciate the improvement in clarity that the reviewer's suggestion generates over the text.

And just one minor comment:

- Revise the text for clarity and flow. In particular, the introduction feels unnecessarily wordy, and the description of the PANGO system could be left out.

The PANGO classification system explanation has been omitted from the introduction, as we believe this revised version is more reader-friendly and incorporates key concepts necessary to comprehend the motivation behind our work. We express our gratitude to the reviewer for their insightful observation and their dedicated effort to enhance our manuscript.

July 5, 2023

Prof. Germán Burgos Figueroa
Universidad de Las Americas
Facultad de Medicina
Campus UdlaPark
Vía Antigua a Nayón
Quito, Pichincha 170503
Ecuador

Re: Spectrum05064-22R1 (A tool for the cheap and rapid screening of SARS-CoV-2 variants of concern (VoC) by Sanger sequencing)

Dear Prof. Germán Burgos Figueroa:

It is my pleasure to inform you that your manuscript has been accepted, and I am forwarding it to the ASM Journals Department for publication. You will be notified when your proofs are ready to be viewed.

Sincerely,

Eleanor Powell
Editor, Microbiology Spectrum
